# DETECTING DEEPFAKES WITHOUT SEEING ANY

## ABSTRACT

Deepfake attacks, malicious manipulation of media containing people, are a serious concern for society. Conventional deepfake detection methods train supervised classifiers to distinguish real media from previously encountered deepfakes. Such techniques can only detect deepfakes similar to those previously seen, but not zero-day (previously unseen) attack types. As current deepfake generation techniques are changing at a breathtaking pace, new attack types are proposed frequently, making this a major issue. Our main observations are that: i) in many effective deepfake attacks, the fake media must be accompanied by false facts i.e. claims about the identity, speech, motion, or appearance of the person. For instance, when impersonating Obama, the attacker explicitly or implicitly claims that the fake media show Obama; ii) current generative techniques cannot perfectly synthesize the false facts claimed by the attacker. We therefore introduce the concept of "fact checking", adapted from fake news detection, for detecting zero-day deepfake attacks. Fact checking verifies that the claimed facts (e.g. identity is Obama), agree with the observed media (e.g. is the face really Obama's?), and thus can differentiate between real and fake media. Consequently, we introduce FACTOR, a practical recipe for deepfake fact checking and demonstrate its power in critical attack settings: face swapping and audio-visual synthesis. Although it is training-free, relies exclusively on off-the-shelf features, is very easy to implement, and does not see any deepfakes, it achieves better than state-of-the-art accuracy.

## 1 INTRODUCTION

> *The ability to disseminate large-scale disinformation to undermine scientifically established facts poses an existential risk to humanity and endangers democratic institutions and fundamental human rights.*
>
> —— UN Report

Deepfakes have been universally acknowledged to pose a grave threat to society. Bad actors can use fake information for various malicious purposes, including disinformation, societal polarization, embarrassment, and privacy violations. Detecting and flagging deepfakes can be an effective way of overcoming this threat. The astonishing rise in quality and prevalence of deep generative models allows fast and credible deepfake production, making it essential to develop automatic means for detecting them. This challenge has been taken up by the machine learning community, which has developed many detection methods. Despite significant progress, current methods are insufficient for tackling this formidable challenge. For example, approaches based on detecting high-frequency artifacts created by generative models, can be overcome by removing the artifacts or slightly changing model architectures. Today's most effective approaches use a supervised classifier trained to discriminate between real images and previously seen deepfake media. These methods assume that future deepfake attacks will be similar to previously observed ones, thus often do not generalize well to previously-unseen "zero-day[1]" deepfake attacks. This problem will only get worse, as generative models, which are the key technology behind deepfakes, are advancing at an unprecedented pace. There is therefore a clear and immediate need for new methods that overcome this generalization gap.

This paper suggests using the concept of *fact checking* for detecting zero-day deepfake attacks. This idea is adapted from fake news detection (Hassan et al., 2015; Rashkin et al., 2017; Guo et al., 2022).

---

[1]Zero-day attack: An attack that exploits a previously unknown vulnerability

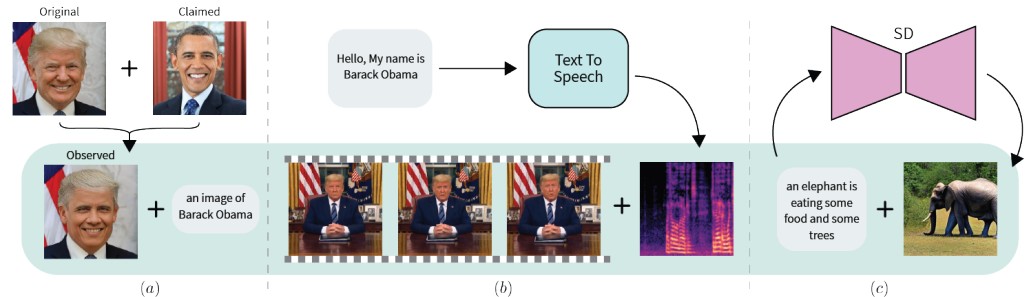

Figure 1: False facts in deepfake attacks. *(a)* Face forgery: the claimed identity is seamlessly blended into the original image. The observed image is accompanied by a false fact i.e., "an image of Barack Obama". *(b)* Audio-Visual (AV): fake audio is generated to align with the original video or fake video is generated to align with the original audio. Fake media are accompanied by a false fact, that the video and audio describe the same event. *(c)* Text-to-Image (TTI): the textual prompt is used by a generative model e.g. Stable Diffusion, to generate a corresponding image. The fake image is accompanied by a false fact, that the caption and the image describe the same content.

In some of the most important deepfake attack scenarios, the fake media (image, video, etc.) must be accompanied by false facts e.g. claims about facial identity, speech, activity or appearance. Here, we deal with several critical scenarios illustrated in Fig. 1: i) face manipulation attacks, wherein the identity in a video is manipulated to that of the impersonated person. This is usually accompanied by a fact, namely, a claim about the identity of this person, e.g., "A video of Barack Obama playing the harp"; ii) audio-visual attacks, where a video is manipulated to appear as if a person utters some speech or the audio is manipulated to align with some video. The claimed fact is that the video and audio describe the same event; iii) we explore the extension of this concept beyond facial attacks, analyzing a simplified text-to-image attack scenario where the fake image is augmented with a prompt. The false fact is that the fake image and prompt describe the same content.

The key assumption made in this work is that current generative models are not yet accurate enough to encode the false fact into fake media with sufficient accuracy. For example, when manipulating the facial identity in a video, the observed fake identity will be distinguishable from the identity claimed by the attacker. By checking the false fact provided by the attacker (claimed facial identity), we can distinguish between real and fake images and videos. For instance, we can detect fake images when they do not match Obama's facial identity. We present FACTOR, a general recipe for implementing fact checking, illustrated in Fig. 2. In our formulation, false facts assert that the fake media describe the same content as some other media. FACTOR computes the *truth score* (similarity function) between the media. In all cases shown here, we use off-the-shelf features pretrained on real data. If the truth score is low, we predict that the fact is false, and the media are fake sources of information.

Despite being training-free and very simple to implement, not using any fake data for pretraining and relying entirely on off-the-shelf features, we demonstrate the superiority of our approach across many competitive benchmarks. *Our main contributions are:*

1. Proposing the concept of fact checking for deepfake detection.
2. Introducing FACTOR, a practical recipe for deepfake fact checking and demonstrating its power in critical attack settings: face swapping and audio-visual synthesis.
3. Exploring the applicability of fact checking beyond facial attacks.

## 2 RELATED WORK

**Image synthesis.** Fake images are created either by manipulating parts of existing images or by generating them from scratch. Examples of the former include techniques that modify attributes in a source image or those that replace the original face in an image or video with a target face (Korshunova et al., 2017; Bao et al., 2018; Perov et al., 2020; Nirkin et al., 2019). The other class of methods, however, involves generating all pixels from scratch, whether from random noise (Karras et al., 2019) or text prompts (Rombach et al., 2022; Ramesh et al., 2022; Saharia et al., 2022).

**General deepfake detection.** As deepfake technology advances, significant efforts have been devoted to identifying manipulated media. Traditional approaches focus on examining image statistics changes, detecting cues such as compression artifacts (Agarwal & Farid, 2017). Learning-based methods have also been employed, with an initial emphasis on whether classifiers could effectively distinguish images from the same generative model (Wang et al., 2019; Frank et al., 2020; Rössler et al., 2019). Recent studies (Wang et al., 2020; Chai et al., 2020) shift towards classifiers capable of generalizing to different generative models, demonstrating the efficacy of neural networks trained on real and fake images from one GAN model for detecting images from other GAN models. However, Ojha et al. (2023) emphasizes the non-generalizability of neural networks to unknown families of generative models when trained for fake image detection.

**Face forgery detection.** Early approaches relied on supervised learning to transform cropped face images into feature vectors for binary classification (Dang et al., 2020; Nguyen et al., 2019; Rössler et al., 2019). However, it became evident that relying solely on classification methods had limitations, often leading to overfitting of training data and potentially missing subtle distinctions between real and fake images. Incorporating frequency information proved invaluable for face forgery detection, enabling the identification of specific artifacts associated with manipulation (Frank et al., 2020; Li et al., 2021; Qian et al., 2020; Luo et al., 2021; Liu et al., 2021a). However, it is noteworthy that these cues can sometimes be overcome by techniques such as artifact removal or slight alterations to model architectures. Recent research efforts have increasingly prioritized the improvement of generalization in forgery detection models, recognizing the significance of detecting previously unseen forgeries (Cao et al., 2022; Sun et al., 2022; Zhuang et al., 2022). Huang et al. (2023) uses face identity and face recognition features for supervised training of deepfake detectors. In contrast, our approach does not need training, and therefore enjoys better generalization.

**Audio-Visual deepfake detection.** In the domain of identifying manipulated speech videos, prior research focused on exploiting audio-visual inconsistencies as a crucial cue. Many approaches, rooted in supervised learning, have been devised to directly train audio-visual networks, enabling them to discern video authenticity (Chugh et al., 2020; Mittal et al., 2020). Recently, attention has shifted towards audio-visual self-supervision as a pretraining strategy. This entails self-supervised training, followed by fine-tuning with real/fake labels (Grill et al., 2020; Haliassos et al., 2022). Some methods incorporate lip-reading data for this purpose (Haliassos et al., 2021), while others implicitly integrate it into audio-visual synchronization signals (Zhou & Lim, 2021). Feng et al. (2023) proposed AVAD, probably the most related method, which adapts ideas from anomaly detection for AV deepfakes, and does not use fake data for training. The method requires training a multimodal transformer using multi-objective terms. Our method achieves higher accuracy while being far simpler, using only off-the-shelf feature extractors and not requiring training.

## 3 Fact Checking for Deepfake Detection

This paper introduces the concept of fact checking for detecting deepfake attacks. Previous approaches train supervised classifiers to discriminate real from previously seen deepfake data. This is an effective approach for detecting attacks similar to those seen before, but it does not generalize well to zero-day attacks. Instead, we introduce the concept of fact checking for deepfake detection, which we adapt from fake news detection (Hassan et al., 2015; Rashkin et al., 2017; Guo et al., 2022). Many effective deepfake attacks are accompanied by false facts; facts are claims about the fake media such as their facial identity, speech, actions or appearance. Claiming these facts is essentially the core of the attack. For example, a face swapping attack must be joined with a claim such as "the identity of the person in this image is Obama". This can be stated in the caption provided by the attacker e.g. "President Obama smiling", or it is implicitly inferred from the context.

Due to inherent imperfections in generative models, the false facts are not seamlessly embedded within the deepfake; for example, the manipulated face does not precisely correspond to Obama. This motivates the idea of fact checking for deepfake detection. We determine if the media are fake by verifying if the claimed facts are false e.g. the face does not belong to Obama. As fact checking models can be based entirely on real media, they do not depend on which deepfakes were previously observed. This gives fact checking the ability to generalize to arbitrary deepfake attacks.

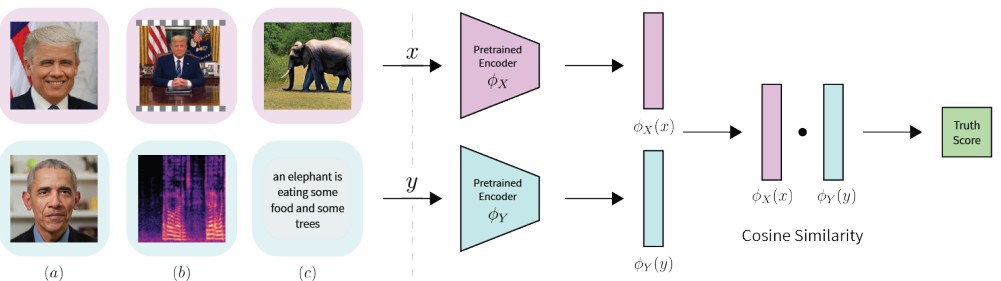

Figure 2: Illustration of FACTOR, our proposed deepfake detection method. FACTOR leverages the discrepancy between false facts and their imperfect synthesis within deepfakes. By quantifying the similarity using the truth score, computed via cosine similarity, FACTOR effectively distinguishes between real and fake media, enabling robust detection of zero-day deepfake attacks.

## 4  FACTOR: A PRACTICAL RECIPE FOR FACT CHECKING

We propose a practical recipe for implementing fact checking for deepfake detection. We formulate facts as statements that can be parsed as "$x$ describes the same content as $y$". This simple formulation covers many interesting cases such as: i) "image $x$ has identity $f$" which can be parsed to "image $x$ describes the same identity as image $y$: for every image $y$ with true identity $f$"; ii) "the video $x$ depicts the same event as the audio $y$"; iii) "the content in image $x$ is being described by the text caption $y$".

We first obtain off-the-shelf encoders for $x$ and $y$ denoted by $\phi_X, \phi_Y$. The encoders are required to return similar values when $x$ and $y$ describe the same content and distinct values otherwise. We measure the similarity using the cosine similarity, name it the *truth score* and denote it by $s$:

$$s(x, y) = \frac{\phi_X(x) \cdot \phi_Y(y)}{\|\phi_X(x)\|_2 \cdot \|\phi_Y(y)\|_2} \qquad (1)$$

Claimed facts with low truth scores are classified as false. The encoders can be obtained off-the-shelf (as is the case for all experiments in this paper) and their pretraining procedure does not require any fake images, rendering implementation remarkably straightforward. We illustrate FACTOR in Fig. 2.

FACTOR improves deepfake detection by effectively addressing zero-day attacks. In the following sections, we showcase the effectiveness of our method in key deepfake scenarios, providing comprehensive analysis and validation.

## 5  FACE SWAPPING DETECTION

Face forgery detection involves identifying instances of manipulated facial features. This task has significant real-world implications, as most deepfakes involve human faces. Face swapping replaces the original face in an image or video with the claimed identity. The ultimate goal is to generate a fake face that is indistinguishable from a real one to the human eye.

### 5.1  METHOD

Our approach relies on a key observation: in most face swapping attacks, the attacker states the claimed identity of the generated face as a (false) fact e.g. "Barack Obama smiling". Due to the limitations of current generative methods, the claimed identity is not perfectly transferred to the fake image. We thus propose to distinguish between real and fake images by verifying that the facial identity claimed by the user matches that of the observed image. We assume the availability of a reference set (with as few as a single image) containing authentic facial images of the claimed identity. Such a set of images is very easy to obtain; for example by downloading from the web or by asking the attacked identity looking to clear their name.

Our method takes as input a face image $x$ and its claimed identity $f$. We also require a face recognition model, denoted by $\phi_{id}(.)$, to compute facial features (we use the open-source model

Table 1: Performance comparison (average ROC-AUC %) of baseline methods on the DFDC dataset, evaluating their ability to detect zero-day deepfake attacks. Our method outperforms supervised baselines, underscoring its robustness to previously unseen manipulation techniques. A and B are two distinct deepfake generation methods.

| Scenario | Train/Test | Xception | EffNetB4 | FFD | F3NET | SPSL | RECCE | UCF | Ours |
|----------|-----------|----------|----------|-----|-------|------|-------|-----|------|
| In-Method | A/A | 95.7 | 83.5 | 96.0 | 97.4 | 96.4 | 95.0 | 97.0 | **99.9** |
| | B/B | 93.2 | 64.4 | 85.7 | 89.4 | 81.7 | 88.6 | 94.9 | **100.** |
| | Mean | 94.5 | 74.0 | 90.9 | 93.4 | 89.1 | 91.8 | 96.0 | **99.9** |
| Zero-Day | A/B | 74.0 | 70.7 | 77.9 | 79.8 | 84.4 | 74.2 | 81.3 | **100.** |
| | B/A | 65.9 | 41.6 | 56.6 | 87.9 | 48.7 | 62.0 | 67.1 | **99.9** |
| | Mean | 70.0 | 56.2 | 67.3 | 83.9 | 66.6 | 68.1 | 74.2 | **99.9** |

Table 2: Performance comparison of baseline methods and our fact checking based FACTOR. The supervised models were trained on FF++(C23) and evaluated on Celeb-DF, DFD, and DFDC datasets.

| Dataset | Cross-Dataset | | | | | | | Ref. Set |
|---------|----------|----------|-----|-------|------|-------|-----|----------|
| | Xception | EffNetB4 | FFD | F3NET | SPSL | RECCE | UCF | Ours |
| Celeb-DF | 73.7 | 73.9 | 74.4 | 73.5 | 76.5 | 73.2 | 75.3 | 97.0 |
| DFD | 81.6 | 81.5 | 80.2 | 79.8 | 81.2 | 81.2 | 80.7 | 96.3 |
| DFDC | 73.7 | 72.8 | 74.3 | 73.5 | 74.1 | 74.2 | 75.9 | 99.7 |
| Mean | 76.3 | 76.1 | 76.3 | 75.6 | 77.3 | 76.2 | 77.3 | 97.7 |

Wang et al. (2017)). We construct a reference set $R_f$ consisting of real images of the stated identity $f$. Subsequently, we measure the similarity between the test image $x$ and each image within our reference set using cosine similarity over $\phi_{id}(.)$ features. Note that in this scenario, both $x$ and $y \in R_f$ are facial images, thus $\phi_{id} = \phi_X = \phi_Y$. The truth score $s(x)$ of the image $x$ is the similarity to the nearest face in the reference set. Low truth scores indicate that the fact is false and the image is fake. Formally, the truth score is given by:

$$s(x) = \max_{y \in R_f} \{sim(\phi_{id}(x), \phi_{id}(y))\}$$

## 5.2 EXPERIMENTS

**Datasets.** We conducted experiments on three face swapping datasets that provide identity-related information: Celeb-DF (Li et al., 2020), DFD (Research et al.), and DFDC (Dolhansky et al., 2019). Other standard face swapping datasets do not include identity information. The Celeb-DF dataset was generated through face swapping involving 59 pairs of distinct identities, comprising 590 real videos and 5,639 fake videos. DFD, on the other hand, is a deepfake dataset characterized by 363 real videos and 3,068 synthetically generated fake videos. The DFDC dataset stands out as the largest publicly available collection of face-swapped videos, featuring 1,133 real videos and 4,080 manipulated videos for testing. This dataset poses a substantial challenge for existing forgery detection methods, due to the diverse and previously unseen manipulation techniques it contains. To ensure uniformity in our experiments, we uniformly subsampled each video (train or test and for all datasets) into 32 frames. Furthermore, we split each identity's authentic videos into a 50/50 train-test split. FACTOR uses the 50% subset of training videos as its reference set (it does not require training), denoted by $R_f$. In contrast, the supervised baselines use this subset exclusively for training their models. The test set for claimed identity $f$ consisted of 50% of the real videos of this identity and fake video clips of with $f$ being the claimed identity. We followed the literature in using frame-level evaluation, computing ROC-AUC scores across all frames. Our reported results are an average of performance computed for all identities within the dataset. Full implementation details are in Appendix A.1.

**Results.** In order to assess the robustness of our proposed method against zero-day deepfake attacks, we conducted experiments on the DFDC dataset, which contains real and fake images. The fakes were created by two distinct deepfake generation methods. Method A and method B denote the deepfake generation methods within the DFDC dataset. Note that the dataset does not provide specific information about the technical details of these deepfake generation techniques; they are simply

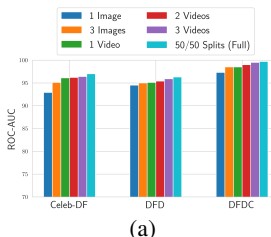 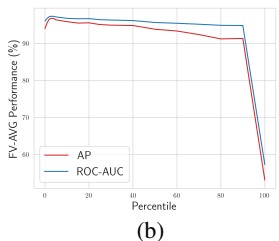 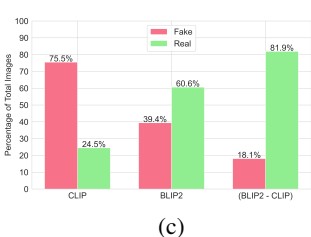

Figure 3: (*a*) Ablation study over our reference set size (ROC-AUC %). Even a single image reference set results in performance close to the full set performance. (*b*) Ablation study on the percentile $\lambda$ for truth score selection. Performance is robust across a wide range of values. (*c*) Comparison of truth scores for different text-image encoders. We report the percentage of image captions whose fake images had a higher truth score than the original image and vice versa for CLIP, BLIP2, and BLIP2-CLIP. We can see that CLIP and BLIP2 truth scores have opposite effects while the proposed rule BLIP2-CLIP is the most effective.

identifiers within the DFDC dataset metadata. To establish a comparative baseline, we selected a range of classic and contemporary state-of-the-art methods, including Xception (Rössler et al., 2019), EfficientNetB4 (Tan & Le, 2019), FFD (Dang et al., 2020), F3Net (Qian et al., 2020), SPSL (Liu et al., 2021a), RECCE (Cao et al., 2022), and UCF (Yan et al., 2023). Each baseline model was trained to classify real vs. deepfakes generated by method A and subsequently evaluated on real images vs. method B or vice versa. This ensured that no baseline model had prior exposure to the test-time deepfake generation method. In contrast, our encoders were exclusively pretrained on real data and not on fake data from methods A or B. The results, presented in Tab. 1, show that while the supervised baselines performed poorly on zero-day attack scenarios, our method achieved near-perfect accuracy. Additionally, to showcase the generalization challenges of supervised methods, we also tested them on real vs. fake data from the *same* method as for training (e.g. A). We can see in Tab. 1 that the baselines perform much better in this case. Note that our method outperforms even in this case, as DFDC does not suffer from serious artifacts making it challenging for methods that use visual artifacts rather than identity.

To simulate a common zero-day setting, where the dataset does not have previously observed deepfake attack data, we compared our method against state-of-the-art approaches that are trained on a large external dataset (here, FF++(C23) (Rössler et al., 2019)), and evaluated on the zero-day attack data in another dataset (here evaluated on Celeb-DF, DFD, and DFDC). As our method does not require training, we only used a reference set of real images from the claimed identity. We did not need to train on FF++. The results can be seen in Tab. 2, our method is far more effective on such zero-day attacks. It is clear that supervised methods struggle to generalize well across both datasets and attack types. Fact checking removes this strong requirement for in-method, in-dataset training data, resulting in improved performance.

**Ablation.** We conducted sensitivity analysis on the effect of reference set size on performance. The results in Fig. 3a demonstrate the method's robustness to reference set variations. A minimal reference set containing only a single image results in only a slight decrease in accuracy. Note that videos were uniformly subsampled to 32 frames in our experiments.

**Limitations.** i) If an attacker simply copies the claimed face onto the observed image, it will correspond to the claimed face identity, although this would result in an unrealistic appearance. To mitigate this, we recommend ensembling our method with a simple image realism-based approach which will easily catch such crude attacks. ii) Our method does not deal with cases where the original and claimed identities are identical, but other attributes are manipulated, such as changes in facial expressions, age or other non-identity-related features. These tasks are left for future work.

Table 3: AP and AUC (%) FakeAVCeleb results, following the AVAD (Feng et al., 2023) evaluation protocol. Supervised methods are evaluated on unseen fake types. Best results are in bold.

| | Method | Mode | Pretrained Dataset | Category | | | | | | | | | | | |
| | | | | RVFA | | FVRA-WL | | FVFA-WL | | FVFA-FS | | FVFA-GAN | | AVG-FV | |
| | | | | AP | AUC | AP | AUC | AP | AUC | AP | AUC | AP | AUC | AP | AUC |
| Supervised | Xception | $\mathcal{V}$ | ImageNet | – | – | 88.2 | 88.3 | 92.3 | 93.5 | 67.6 | 68.5 | 91.0 | 91.0 | 84.8 | 85.3 |
| | LipForensics | $\mathcal{V}$ | LRW | – | – | **97.8** | **97.7** | 99.9 | 99.9 | 61.5 | 68.1 | 98.6 | 98.7 | 89.4 | 91.1 |
| | AD DFD | $\mathcal{AV}$ | Kinetics | **74.9** | **73.3** | 97.0 | 97.4 | 99.6 | 99.7 | 58.4 | 55.4 | **100.** | **100.** | 88.8 | 88.1 |
| | FTCN | $\mathcal{V}$ | – | – | – | 96.2 | 97.4 | **100.** | **100.** | 77.4 | 78.3 | 95.6 | 96.5 | 92.3 | 93.1 |
| | RealForensics | $\mathcal{V}$ | LRW | – | – | 88.8 | 93.0 | 99.3 | 99.1 | **99.8** | **99.8** | 93.4 | 96.7 | **95.3** | **97.1** |
| Unsupervised | AVBYOL | $\mathcal{AV}$ | LRW | 50.0 | 50.0 | 73.4 | 61.3 | 88.7 | 80.8 | 60.2 | 33.8 | 73.2 | 61.0 | 73.9 | 59.2 |
| | VQ-GAN | $\mathcal{V}$ | LRS2 | - | - | 50.3 | 49.3 | 57.5 | 53.0 | 49.6 | 48.0 | 62.4 | 56.9 | 55.0 | 51.8 |
| | AVAD | $\mathcal{AV}$ | LRS2 | 62.4 | 71.6 | 93.6 | 93.7 | 95.3 | 95.8 | 94.1 | 94.3 | 93.8 | 94.1 | 94.2 | 94.5 |
| | AVAD | $\mathcal{AV}$ | LRS3 | 70.7 | 80.5 | 91.1 | 93.0 | 91.0 | 92.3 | 91.6 | 92.7 | 91.4 | 93.1 | 91.3 | 92.8 |
| | Ours | $\mathcal{AV}$ | LRS3 | **98.6** | **98.7** | 94.4 | 95.7 | 97.4 | 97.7 | 97.8 | 98.1 | 97.6 | 97.9 | **96.8** | **97.4** |

# 6 AUDIO-VISUAL DEEPFAKE DETECTION

## 6.1 METHOD

Audio-visual (AV) deepfakes involve synthesizing video to match given audio, audio to match given video, or synthesizing both. AV deepfakes are typically also accompanied by a false fact about the claimed identity of the speaker, which can be dealt with using the tools of Sec. 5. Here, we focus on another false fact, the fact that the two media indeed correspond to the same event. Audio-visual synthesis is very hard for current generative models and we expect them to achieve only partial results. We adapt our method, FACTOR, to AV data by using powerful off-the-shelf audio-visual encoders (we use AV-Hubert (Shi et al., 2022)) to extract features for each modality. The audio and video encoders are denoted by $\phi_A$ and $\phi_V$, respectively. We then use the cosine similarity between them as the truth score. There is an added complication in this case, as AV deepfakes are scored at a video level, while the truth score is calculated for every temporal frame. We opt for a simple but effective solution, using the truth score with the $\lambda\%$ lowest value in the video (we choose $\lambda = 3\%$, but a wide range of values is successful, see Fig. 3b). Formally, for a clip of length $T$, we denote the visual frame at time $t$ by $v_t$ and the audio frame by $a_t$. The truth score for frame $t$ (denoted $s_t$) is given by:

$$s_t = sim(\phi_V(v_t), \phi_A(a_t)) \tag{2}$$

Where $sim$ denotes cosine similarity. We choose the frame value with the $\lambda\%$ percentile as the overall clip truth score $s$. This is given by:

$$s = perc(\{s_1, s_2..s_T\}, \lambda) \tag{3}$$

Where $perc(\cdot, \lambda)$ calculates the $\lambda\%$ percentile of the set. AV data with some misaligned frames will obtain a low truth score indicating a high likelihood of being fake. Real data will not have mismatches and will achieve high truth scores. Note that misalignment between audio-visual data has been detected by several previous deepfake detection methods including Feng et al. (2023). The novelty here lies in demonstrating that our universal fact checking approach outperforms the previous methods, using a simpler, streamlined method.

## 6.2 EXPERIMENTS

**Datasets.** We evaluated our method on the FakeAVCeleb video forensics dataset (Khalid et al., 2021). This dataset contains a diverse range of manipulations that alter both human speakers' speech and facial features, reflecting real-world deepfake scenarios. Specifically, FakeAVCeleb is derived from the VoxCeleb2 dataset and consists of 500 authentic videos, and 19,500 manipulated videos. These manipulations are generated through various techniques, including Faceswap (FaceSwap.), FSGAN (Nirkin et al., 2019), Wav2Lip (Prajwal et al., 2020), and the incorporation of synthetic sounds generated by SV2TTS (Jia et al., 2018). The dataset features examples that exhibit different combinations of these manipulations, capturing the diverse nature of deepfake content. Full implementation details and further evaluation can be found at Appendix A.2.

**Settings and baselines.** We conducted experiments on the FakeAVCeleb dataset (Khalid et al., 2021) following the protocol established by AVAD (Feng et al., 2023). We report numbers by SOTA

supervised methods: Xception (Rössler et al., 2019), LipForensics (Haliassos et al., 2021), AD DFD (Zhou & Lim, 2021), FTCN (Zheng et al., 2021), and RealForensics (Haliassos et al., 2022). We also compared to other self-supervised methods: AVBYOL (Grill et al., 2020; Haliassos et al., 2022), VQGAN (Esser et al., 2021) and AVAD (Feng et al., 2023). We use the same categorization of the FakeAVCeleb dataset as in Feng et al. (2023): (i) RVFA: real video with fake audio by SV2TTS; (ii) FVRA-WL: fake video by Wav2Lip with real audio; (iii) FVFA-WL: fake video by Wav2Lip, and fake audio by SV2TTS; (iv) FVFA-FS: fake video by Faceswap and Wav2Lip, and fake audio by SV2TTS; (v) FVFA-GAN: fake video by FSGAN and Wav2Lip, and fake audio by SV2TTS. For supervised methods, we omitted the evaluated category during training and used the remaining ones.

**Results.** The results presented in Tab. 3 underscore the superior performance of our method across all categories, surpassing self-supervised approaches (AVBYOL, VQGAN, and AVAD) by a significant margin. Our method consistently demonstrates comparable or superior performance to supervised methods in all categories, despite not relying on labeled supervision or fake data. Notably, our method outperforms all supervised baselines in terms of average AP and ROC-AUC. Although supervised baselines excel in certain categories, their performance deteriorates in others demonstrating poor generalization skills. This highlights the robustness and effectiveness of our method for identifying fake videos manipulated by diverse and previously unseen zero-day attacks. A further evaluation of FACTOR on the KoDF (Kwon et al., 2021) dataset is provided in Appendix A.2.

**Ablation.** We ablate the effect of the percentile $\lambda$ in Fig. 3b. The results are robust to this hyper-paramer, with a mere 2% decrease in fake video average performance when comparing $\lambda = 0 - 90\%$.

## 7 ANALYSIS: OVERFITTING EFFECTS

Here we analyse the effect of overfitting on factor by studying a simplified setting, where the attacker first synthesizes a fake image using a text prompt and a text-to-image (TTI) model. The attacker then presents the fake image with the input prompt as its caption. In this case, the false fact is that the text prompt and the image describe the same content. As TTI models are known not to be perfectly aligned with the input prompt (Xu et al., 2023; Chefer et al., 2023), fact checking can be used to detect fake images generated by them. We note that this setting is simplified, as the attacker can choose to use another caption to describe the fake image which differs from the input prompt. This caption can be generated post-hoc (after the fake image was synthesized), making this caption potentially very accurate and a true fact. However, we choose to analyze the case of the caption being identical to the initial prompt as it showcases the interesting overfitting behavior of fact checking methods.

**Dataset.** Our evaluation is performed on a random sample of 1000 images from the COCO (Lin et al., 2014) dataset. Each image has 5 corresponding captions written by different people. We use the popular TTI model Stable Diffusion (SD) to generate an image for each caption. This yields 5000 synthetic images and 1000 real images.

**Truth score paradox.** We begin by using the CLIP (Radford et al., 2021) encoders to encode the caption and image respectively; the truth score is the cosine similarity between them. We compute truth scores for all real and fake images in COCO. We begin with a simple analysis - we compare the CLIP truth scores for each real image with its fake counterparts (fake images with the same caption). We denote classification accuracy, as the number of fake images whose truth score was lower than that of their real counterparts. The results are shown in Fig. 3c. Surprisingly, we see that this method's deepfake detection accuracy is lower than chance! To further explore this phenomenon, we perform the same experiment, but now using BLIP2 (Li et al., 2023) as the text and image encoder. This experiment results are as expected; real images have higher truth scores than their fake counterparts. Full implementation details are in Appendix A.3.

**Resolution of the paradox.** Why do BLIP2 truth scores behave as expected, but CLIP scores do exactly the opposite? We recall that SD was trained with CLIP text features, but not with BLIP2 features. We therefore hypothesize that SD overfits to CLIP scores, making the fake images better aligned with their input caption than real images. On the other hand, the generated image does not fully correspond to the input prompt, due to imperfections in the TTI model. Therefore, an objective multi-modal encoder, i.e. one not used for training the TTI, is able to perform fact checking and identify that fake images do not fit with the claimed prompts. This explains why CLIP truth scores support the claimed fact, but BLIP2 rejects it.

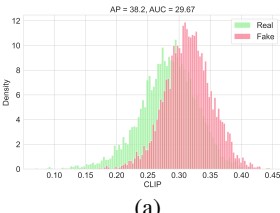 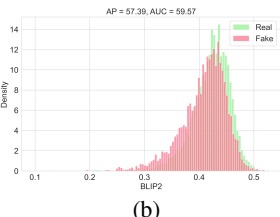 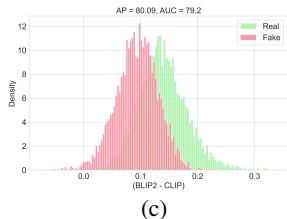

Figure 4: Comparison of truth scores distributions of real images and fake images with respect to the claimed caption using (*a*) CLIP, (*b*) BLIP2, and (*c*) $BLIP2 - CLIP$ scoring rules. (*a*) CLIP truth scores are surprisingly higher for fake images than for real ones. (*b*) BLIP2 truth scores achieve weak separation between real and fake data. (*c*) $BLIP2 - CLIP$ truth scores achieve a stronger separation between real and fake data.

**Method.** We compute the CLIP and BLIP2 truth score for each image and prompt pair $(x, y)$. The final score is the BLIP2 minus the CLIP scores:

$$s(x, y) = sim(\phi_X^{BLIP2}(x), \phi_Y^{BLIP2}(y)) - sim(\phi_X^{CLIP}(x), \phi_Y^{CLIP}(y)) \qquad (4)$$

**Results.** We present the truth score histograms for CLIP, BLIP2, and BLIP2-CLIP in Fig. 4. We find that the CLIP truth score is negative correlated with deepfakes, and BLIP2 is positively correlated. Using BLIP2 - CLIP achieves the best of both worlds. Numerically, CLIP truth score achieves around 30% ROC-AUC, while BLIP2 obtains around 60%. Using the difference between the truth scores yields a much better result of 79.2%. While we do not claim that this setting is realistic (as attackers may not disclose the exact input prompt) or that these results are better than the state-of-the-art on COCO, this scenario demonstrates how overfitting may play an important part in fact checking.

## 8    DISCUSSION AND LIMITATIONS

**Facts must be falsifiable.** As our method detects deepfakes by detecting false facts, it is necessary that the false fact will be falsifiable. For example, a fact that carries no information will not help deepfake detection e.g. the tautological prompt "An image". On the other hand, facial identity, which can distinguish one person out of a billion is certainly helpful. We conducted an investigation of the effect of fact information complexity vs. detection accuracy in Appendix A.4. The results showed that higher fact informativeness resulted in higher detection accuracy.

**Unconditional deepfakes do not include facts.** Our method is not designed for unconditional deepfakes, e.g. a generated image without added information such as a caption, as there are no attacker provided facts to be falsified. We stress that face swapping and audio-visual deepfakes are of sufficient practical and important to make our approach valuable.

**Supervised approaches work well on previously seen attacks.** The primary benefit of our approach is generalizing to unseen, zero-day attacks. Existing supervised techniques are effective on attacks similar to those seen before, for which sufficient training data can be obtained. We note that in some cases, our method outperforms supervised techniques even for previously seen attacks.

**No pretrained encoders for non-standard facts.** FACTOR used off-the-shelf feature encoders to do fact checking in important attack scenarios. It is possible that other facts may require specialized encoders that are not available off-the-shelf. In this case, the user would need to train the encoders rather than using off-the-shelf ones. Still, this will not require using any fake data.

**Hazards of future progress.** To overcome FACTOR, generative models must synthesize false facts significantly better than is currently possible. Specifically, they would have to replicate not only the visual appearance but also the finer details, nuances, and contextual cues of the claimed facts. When generative methods indeed progress to this level, our method would need to be re-evaluated.

## 9    CONCLUSION

This paper proposes the concept of fact checking to address the challenge of detecting unseen, zero-day attacks. We propose FACTOR for implementing this, and showcase it in three important settings.

FACTOR outperforms the state-of-the-art without seeing any fake data, using only pretrained feature encoders and being simple to implement.

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

# Appendix

## Detecting Deepfakes Without Seeing Any

## A   EXPERIMENTAL DETAILS & ANALYSIS

### A.1   FACE SWAPPING DETECTION

**Implementation details.** Our implementation relies on a pretrained Attention-92(MX) model from Face-X-Zoo (Wang et al., 2021), denoted by $\phi_{id}(.)$, originally trained on the MS-Celeb dataset (Guo et al., 2016). To ensure uniformity and consistency in our data, we incorporated robust data processing techniques. These encompassed critical steps such as face detection, precise cropping, and alignment, all facilitated by the DLIB library (Sagonas et al., 2013). The preprocessing ensured that all face images were uniformly cropped and normalized to compact dimensions of $112 \times 112$. We computed the features of all images (observed and referenced) using the above feature encoder and calculated their similarity using cosine similarity. For evaluation, we used Receiver Operating Characteristic Area Under the Curve (ROC-AUC).

**Feature extractor ablation study.** In Tab. 4 we present an ablation study on the effect of $\phi_{id}$. Specifically, we compared our chosen Attention-92(MX) with CLIP (Radford et al., 2021) and Swin-S (Wang et al., 2021; Liu et al., 2021b). The results demonstrate that our method is not overfitted to a single feature extractor. Specifically, we found that CLIP, which has been trained on quite irrelevant data, is already quite effective.

Table 4: Performance comparison (average ROC-AUC %) of different feature extractors.

| Dataset | CLIP | Swin-S | Attention-92(MX) |
|---------|------|--------|------------------|
| Celeb-DF | 89.3 | 94.5 | **97.0** |
| DFD | 95.7 | 96.1 | **96.3** |
| DFDC | 98.8 | 99.3 | **99.9** |

### A.2   AUDIO-VISUAL DEEPFAKE DETECTION

**Implementation details.** In our implementation, we use the AV-HuBERT Large model as our feature encoder. This model was pretrained on real, unlabeled speech videos from the LRS3 dataset. No fake videos at all or any real videos from the evaluation dataset were used in pretraining. We follow the official AV-HuBERT implementation[2] for video preprocessing. Specifically, we use an off-the-shelf landmark detector to identify Regions of Interest (ROIs) within each video clip. Both video and audio components are transformed into feature matrices represented in $\mathbb{R}^{T \times d}$, where $T$ represents the number of frames, and $d$ is the AV-HuBERT feature space dimension ($d = 1024$). Accordingly, we choose $\lambda = 3\%$ for our choice of $\lambda$. However, an ablation study is presented in Fig. 3b which indicates that performance is not sensitive to the choice of $\lambda$. In accordance with standard practice, we used two evaluation metrics: (i) average precision (AP) and (ii) Receiver Operating Characteristic Area Under the Curve (ROC-AUC).

**KoDF Evaluation.** To further assess the cross-domain applicability of our fact checking approach, we conducted an evaluation on the Korean Deepfake Detection (KoDF) dataset (Kwon et al., 2021), following the established protocol outlined by AVAD (Feng et al., 2023). For comparative analysis, we benchmarked our method against several state-of-the-art supervised and self-supervised baselines, including Xception (Rössler et al., 2019), LipForensics (Haliassos et al., 2021), AD DFD (Zhou & Lim, 2021), FTCN (Zheng et al., 2021), VBYOL (Grill et al., 2020; Haliassos et al., 2022), VQGAN (Esser et al., 2021), and AVAD (Feng et al., 2023). The supervised baselines were trained on the FakeAVCeleb dataset (Khalid et al., 2021), which uses similar synthesis techniques to KoDF, such as FaceSwap (FaceSwap.), FS-GAN (Nirkin et al., 2019), and Wav2Lip (Prajwal et al., 2020).

---

[2]https://github.com/facebookresearch/av_hubert

Table 5: AP and AUC (%) KoDF results, following the AVAD (Feng et al., 2023) evaluation protocol. Best results are in bold.

| | Method | Modality | KoDF | |
|---|---|---|---|---|
| | | | AP | AUC |
| Supervised (transfer) | Xception | $\mathcal{V}$ | 76.9 | 77.7 |
| | LipForensics | $\mathcal{V}$ | 89.5 | 86.6 |
| | AD DFD | $\mathcal{AV}$ | 79.6 | 82.1 |
| | FTCN | $\mathcal{V}$ | 66.8 | 68.1 |
| | RealForensics | $\mathcal{V}$ | **95.7** | **93.6** |
| Unsupervised | AVBYOL | $\mathcal{AV}$ | 74.9 | 78.9 |
| | VQ-GAN | $\mathcal{V}$ | 46.8 | 45.5 |
| | AVAD | $\mathcal{AV}$ | 87.6 | 86.9 |
| | Ours | $\mathcal{AV}$ | **92.0** | **93.1** |

The results, summarized in Tab. 5, provide evidence for our method's cross-generalization capability. FACTOR achieves performance levels comparable to many state-of-the-art supervised baselines, and surpasses all unsupervised methods by a large margin. This highlights the adaptability and effectiveness of our method to scenarios with distinct linguistic and cultural attributes.

## A.3 TEXT-TO-IMAGE DEEPFAKE DETECTION

**Implementation details.** In our implementation, we employed two multi-modal feature encoders, CLIP (Radford et al., 2021) and BLIP2 (Li et al., 2023), to encode textual prompts and images. Specifically, for CLIP's architecture, we leveraged the ViT-B/16 pretrained on the LAION-2B dataset, following OPENCLIP specifications (Ilharco et al., 2021). Furthermore, the checkpoint version of Stable Diffusion we used was v1-5 (StableDiffusion.). In order to evaluate the performance of our approach, we used two evaluation metrics: (i) average precision (AP) and (ii) Receiver Operating Characteristic Area Under the Curve (ROC-AUC).

## A.4 CAPTION COMPLEXITY AND TRUTH SCORES

We hypothesize that more complex facts are more falsifiable and therefore improve deepfake detection accuracy. To test this hypothesis, we tested the correlation between textual prompt complexity and their truth scores for real and fake images. We hypothesized that as the complexity of the prompt increased, it would exhibit greater similarity to the real image compared to the fake image. Caption complexity, in this context, refers to the level of detail of describing the image content. To systematically explore this hypothesis, we evaluated a dataset consisting of 1000 randomly selected images from the COCO (Lin et al., 2014) dataset. For each of these real images, we selected two captions: one with the minimum length and another with the maximum length. The minimum length caption represented a simple textual prompt, while the maximum length caption was considered more complex due to its larger word count.

We paired each real image with its corresponding minimum and maximum length captions. For each caption, we generated corresponding fake images, using Stable Diffusion. For each pairing, we calculated truth scores, between the real image and its respective caption, as well as between the fake image and the same caption. This analysis was conducted within the feature spaces of both CLIP (Radford et al., 2021) and BLIP2 (Li et al., 2023), which notably, exhibited disagreements in their prompt similarity trends. Our findings, presented in Fig. 5, underscore a consistent pattern. With increasing complexity of the prompts, as measured through the maximum length captions, more prompts achieved a higher truth score on the real images than on the fakes. This aligns with our hypothesis, demonstrating that more complex prompts contribute to a shift in similarity to the real image, thereby enhancing deepfake detection accuracy. Additionally, we can observe the same phenomenon we witnessed in Sec. 7, wherein CLIP truth scores support the claimed fact (caption), while BLIP2 rejects it. So the effectiveness of BLIP2 truth scores increases with prompt complexity, while using minus CLIP truth scores becomes less effective as prompts become more complex, as the

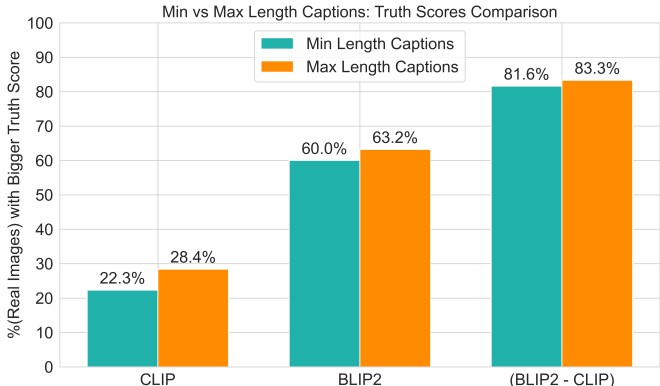

Figure 5: Impact of prompt complexity on truth score. We paired real images with both their minimum and maximum length captions, generating fake versions of those captions. Truth scores were calculated for these pairs. They revealed that as prompt complexity increased, measured through maximum length captions, more prompts achieved higher truth scores with real images, enhancing deepfake detection accuracy. We report the percentage of image captions whose real images had a higher truth score than the fake image for CLIP, BLIP2, and BLIP2-CLIP.

overfitting contrasts with the generative model's failure to synthesize the image corresponding to the complex caption.

