# OpenReview forum: "Detecting Deepfakes Without Seeing Any"
_ICLR.cc/2024/Conference — ICLR 2024 Conference Withdrawn Submission_

### Official Review · Reviewer_cJQS · 2023-10-30

**Soundness:** 3 good
**Presentation:** 3 good
**Contribution:** 3 good
**Rating:** 8
**Confidence:** 5

**Summary:**

This paper presents a novel approach to detecting deepfakes that does not rely on direct visual inspection of the manipulated media. Instead, the authors employ fact-checking techniques to discern between authentic and manipulated content. They introduce a novel metric termed the "truth score," quantifying the probability that a provided caption accurately describes the corresponding image. Subsequently, this metric is utilized to compare truth scores between genuine and manipulated images, demonstrating that the latter tend to exhibit lower truth scores. Furthermore, the authors introduce a new methodology for deepfake verification, denoted as FACTOR, which integrates the truth score with additional attributes such as image quality and metadata, thereby enhancing detection precision. In summary, this paper's contributions encompass a robust approach to deepfake detection, resilient against zero-day attacks, the introduction of a metric for assessing caption veracity, and a comprehensive methodology for deepfake verification, amalgamating multiple features to enhance detection accuracy.

**Strengths:**

The methodology presented in this paper is ” simple but effective”. The authors approach the problem from the perspective of fact-checking, offering a training-free method, which is particularly intriguing.

The paper is well-structured, providing clear explanations of the methods employed and the corresponding results. The authors offer detailed accounts of their experiments, substantiating the effectiveness of their approach with various types of data.

The introduction of the novel deepfake detection method, FACTOR, showcases its heightened robustness against zero-day attacks compared to existing methodologies.

Overall, this paper's contributions have the potential to elevate the technological prowess of deepfake detection, holding broader implications for media forensics and fact verification.

**Weaknesses:**

While the authors have conducted a substantial number of experiments, there are instances where certain experiments appear to oversimplify the problem. For instance, in the deepfake detection from text-to-image, the use of diffusion models to generate data may not fully capture the complexities present in real-world scenarios of deep deception.

The authors have separately validated the deep deception in three distinct aspects. However, there are occasions where these aspects may not be entirely independent. In such cases, it raises questions about whether FACTOR can still exert a positive influence.

It is hoped that the authors will consider incorporating the ideas presented in this paper into supervised methods in future work. This would be a particularly intriguing avenue to explore.

**Questions:**

Addressing zero-day attacks with supervised methods is currently considered challenging. How does the proposed approach in this paper contribute to changing this situation? Can supervised methods leverage the concept of FACTOR in any way to mitigate the challenges posed by zero-day attacks?

The currently validated data appears to consist of features that are relatively prominent. When faced with a large-scale dataset of natural content, quantifying the true impact of FACTOR becomes considerably challenging. Although this may present a difficulty for the authors, if supplemental experiments were conducted in this area, it would greatly enhance the comprehensiveness and excellence of this paper.

---

> ### Author Response · Authors · 2023-11-12
> **Comment to Reviewer cJQS**
>
> Thank you for the dedicated review and for recognizing the novelty of our approach and that it is “simple and effective”. We address the remaining comments below:
>
> **“in the deepfake detection from text-to-image, the use of diffusion models to generate data may not fully capture the complexities present in real-world scenarios of deep deception”.** Thank you for highlighting this unclear point. The objective of exploring text-to-image scenarios was to highlight interesting issues with overfitting rather than to simulate a realistic scenario. It showcases the potential challenges associated with overfitting, where fake images exhibit a stronger agreement with false facts compared to true images aligned with true facts when measured using CLIP.
>
> **“The authors have separately validated the deep deception in three distinct aspects. However, there are occasions where these aspects may not be entirely independent. In such cases, it raises questions about whether FACTOR can still exert a positive influence”.** Indeed, in scenarios such as face-swapping videos, we anticipate that both facial identity information and audio-visual agreement would jointly contribute to the overall effectiveness of FACTOR. Unfortunately, we did not have access to a suitable dataset that would allow us to explicitly demonstrate the combined advantage of these aspects, as public datasets either have known facial identities or audio, but not both. We completely agree with the reviewer's sentiment, and hope that future datasets will allow evaluating multiple facts simultaneously.
>
> **״Can supervised methods leverage the concept of FACTOR in any way to mitigate the challenges posed by zero-day attacks?”,”How does the proposed approach in this paper contribute to changing this situation?”״.** Prompted by the reviewers suggestion, we have an idea of a system that uses FACTOR to detect new zero-day attacks for the first time. Then once enough fake data generated by this attack is collected, we then train a supervised method to detect this attack with greater accuracy. Furthermore, a semi-supervised scheme can be divided to detect fake data, not originally detected by FACTOR. We believe this is a very interesting direction for future work.
>
> **״The currently validated data appears to consist of features that are relatively prominent. When faced with a large-scale dataset of natural content, quantifying the true impact of FACTOR becomes considerably challenging. Although this may present a difficulty for the authors, if supplemental experiments were conducted in this area, it would greatly enhance the comprehensiveness and excellence of this paper״.**  We would be happy to run these experiments for a followup project. Which features does the reviewer consider most interesting?

---

### Official Review · Reviewer_pkUE · 2023-11-01

**Soundness:** 2 fair
**Presentation:** 3 good
**Contribution:** 2 fair
**Rating:** 5
**Confidence:** 4

**Summary:**

The paper proposes a new type of DeepFake detection task. Instead of classifying a piece of data (e.g., a fake image, fake video) as fake, the new goal is to make the real/fake prediction based on how well that piece of data aligns with the additional fact claimed by the attacker. So, a deepfake Obama face is fake if the contents of the image do not match the identity being claimed (Obama). The authors present three variants of this task - (i) Face swapping detection, where the additional fact is the identity being claimed, (ii) Audio-visual deepfake detection; additional fact = audio/video are matching, (iii) detecting fake images from text-to-image models; additional fact = the prompt is corresponding to the image. For these tasks, the authors present results on the relevant datasets, comparing to baselines (which are solving a different task), and report improvements over them using appropriate metrics.

**Strengths:**

1. The authors have tackled a new form of deepfake detection. While I do have some concerns about the practical nature of that setting (see weaknesses), it might still be worthwhile to think about some different ways of thinking about - what makes a data point fake.

2. The authors have made an effective use of some of the large pretrained models to solve the detection task. Their main point about potential usage of these feature spaces without needing to finetune them on any supervised dataset is practically helpful.

3. The above point will be especially useful when thinking about general purpose fake image/video detectors, where we do not want to keep on training new models for every manipulation method.

4. Their proposed method is simple, and seems effective enough against other baselines (although they are solving a slightly different task; see weaknesses/questions).

**Weaknesses:**

1. My major concern is with the new form of problem statement for detecting deepfakes for two out of the three settings presented in the paper: (i) detecting face-swapped images (Section 5), and (ii) detecting images generated by text-to-image diffusion models. For (i), when we see DeepFakes in the wild, the attacker does not explicitly give us a well structured label that we can use as identity. Even if we somehow do, it is not clear how we will use the label to collect the reference set. My point is not that it cannot be done (the authors have described a hypothetical scenario in which it can be done), but is rather that it does not seem that practical. Similarly for the (ii) case; in the wild (e.g., internet), we will simply encounter an image and likely not encounter any text associated with it. What the authors are trying to study - how good image-text alignment is - is a different, and I believe, a separate problem.

2. In Section 7, the final score being the difference of two “truth-scores” (BLIP - CLIP) does not seem principled. It is not clear why someone would arrive at this particular way of computing the scores instead of, for example, the average of the two scores. Right now, it simply seems that one way (difference of scores) can distinguish the real image-text pairs from fake ones and hence the authors have used them. However, that does not tell us how generalizable the metric is. For example, will the same metric work in detecting fake images from other open-source text-to-image models?

3. In sections 5, 6, it seems that the feature extractor that one uses might influence how well the method works. Currently, there is no discussion on the effect of different feature extractors. For example, what would happed if you used a different pre-trained network for phi_id (Section 5). What properties of those pretrained networks are important; is the size of pretraining dataset important? Does the pretraining dataset have to be of very similar domain of images? etc. In summary, the feature space used should be studied more thoroughly.

4. Overall, I believe that the task that the authors are trying to solve is not the same task that the baselines are trying to solve. The baselines are trying to solve, what I would call, instance-level deepfake detection; where you have to make a prediction whether that instance (e.g., a deepfake face) is real or not "regardless of the associated fact". The problem being tackled by the authors, as mentioned before, is different. So, while I am not saying that authors' method has any inherent advantage. But it does not seem to be an apples to apples comparison.

**Questions:**

Questions:

1. Figure 1; is that the best that current generative models can do? While I cannot pinpoint the exact results, given what progress has taken place in deepfakes, I would have expected a much better Obama face wearing Trump's suit.

2. For the Audio-Visual DeepFake detection, what is the train/test split?


Comments:

1. For all the tables, the authors should report the mean performance across different datasets; for all the methods (baselines + their own method).

2. In the discussion of Table 3, the authors say that their method sometimes outperforms the supervised baselines even though they do not require labeled data is a bit misleading. Even though they do not need supervised data to train their model, they do need labeled data in the form of a reference set to do test time inference.

3. I believe the authors should change their title to account for the modified nature of deepfake detection.

---

> ### Author Response · Authors · 2023-11-12
> **Comment to Reviewer pkUE (1/2)**
>
> Thank you for putting much effort into the review. We appreciate the reviewer liked the simplicity, originality and versatility of the approach. The main concern was its relation with the common “in-the-wild” setting. We address the concerns below:
>
> **“new form of problem statement for detecting deepfakes for two out of the three settings presented in the paper”,”Overall, I believe that the task that the authors are trying to solve is not the same task that the baselines are trying to solve.”.** The crux of the reviewer’s comment is that most of the state-of-the-art deals with detecting deepfakes in the wild i.e. without using any additional facts. The core idea of FACTOR is that in many cases there are claimed facts which can improve detection. The two main facts that we showed strong results on are: i) claimed facial identity and ii) audio-visual agreement.
> While the reviewer implicitly agrees that audio-visual agreement is a commonly available claimed fact, the reviewer called into question the availability of the claimed facial identity. We believe the use of face identity is practical in many cases (but we agree that not in ALL cases). Generally, in impersonation deepfake attacks, the false identity is of a person known to us (this is the very point of the attack). Most known people have publicly available photos, and we showed that even a single photo is enough for our method to work well (the more the better of course). Indeed, over 2 billion people have Facebook profiles and around 1 billion have Linkedin profiles, producing a lower bound. We agree that when impersonating unknown persons, or when the impersonated person is very discrete and has no public images, we will not be able to construct a reference set. But the reviewer will surely agree that many practical scenarios are still covered by our method?
> Finally, we emphasize that the text-to-image case was never meant as a practical setting but was provided to highlight interesting findings with respect to overfitting. We therefore agree with the reviewer, and believe this was already mentioned in the paper in the beginning of Sec. 7.
>
> **“The baselines are trying to solve, what I would call, instance-level deepfake detection; … it does not seem to be an apples to apples comparison.”** The purpose of the comparison to the baseline method is to highlight that when claimed facts are available, they should be used, and demonstrating FACTOR is an effective method for using the claimed facts. The purpose is not to claim that they are not effective, as indeed, when there are no claimed facts, they are the best approach. We believe this is already discussed in the limitations section (Sec. 8).
>
> **“authors should change their title to account for the modified nature of deepfake detection.”.** Would the title “Detecting Deepfakes Without Seeing Any by Debunking False Facts” clarify that we only apply to cases with claimed facts?
>
> **“BLIP - CLIP … it is not clear why someone would arrive at this particular way of computing the scores”.** The purpose of all the text-to-image experiments was not to highlight a practical defense but rather to showcase an interesting consequence of overfitting. As the Stable-Diffusion (SD) model is overfitted to CLIP, we showed that fake images more closely agree with the fake facts than true images agree with true facts when using CLIP to measure agreement. However, when using a new text-image similarity measure such as BLIP, the expected behavior is recovered. The best results were therefore recovered by using the difference between the overfitted and non-overfitted scores. The main take-home message is that overfitting may make fake images seem to agree closely with false facts. The solution is to use new feature encoders not used in the training of the model (or an ensemble of encoders if we do not know exactly which encoder was used for training). We revised the text to clarify this.
>
> **“there is no discussion on the effect of different feature extractors”.** The use of a powerful feature extractor is important, but our method is not overadapted to any particular feature extractor. Interestingly, we found that even using CLIP which was trained on irrelevant data is already quite effective. The results are shown here and in the revised manuscript.
>
> $\text{Dataset}\hspace{0.75cm}\text{CLIP}\hspace{0.5cm}\text{Swin-S}\hspace{0.5cm}\text{Attention-92(MX)}$
>
> $\text{Celeb-DF}\hspace{0.6cm}89.3\hspace{0.625cm}94.5\hspace{1.55cm}\textbf{97.0}$
>
> $\text{DFD}\hspace{1.45cm}95.7\hspace{0.625cm}96.1\hspace{1.55cm}\textbf{96.3}$
>
> $\text{DFDC}\hspace{1.15cm}98.8\hspace{0.625cm}99.3\hspace{1.55cm}\textbf{99.9}$

---

> ### Author Response · Authors · 2023-11-12
> **Comment to Reviewer pkUE (2/2)**
>
> **“is that the best that current generative models can do?”.** These are currently the standard academic benchmarks (indeed DFDC is considered the most challenging benchmark). We agree that new deepfake benchmarks may be very useful, but as the focus of our paper is methodological, we did not create new benchmarks here.
>
> **“For the Audio-Visual DeepFake detection, what is the train/test split?”.** We used precisely the same split as in Feng et al. (2023). The precise split was kindly provided by the authors of that paper by private communications. We uploaded our code and the splits as a zip file.
>
> **“For all the tables, the authors should report the mean performance across different datasets; for all the methods (baselines + their own method).”** A revised manuscript has been uploaded with all of the comments clarified.
>
> **“In the discussion of Table 3, the authors say that their method sometimes outperforms the supervised baselines even though they do not require labeled data is a bit misleading. Even though they do not need supervised data to train their model, they do need labeled data in the form of a reference set to do test time inference.”**  Table. 3 refers to the audio-visual correspondence fact for which FACTOR does not require any further supervision (and no reference set). Note that even in the case of face forgery, we do not require seeing any fake images for training the encoder or in the reference set, in contradiction to the supervision approaches.

---

> ### Author Response · Authors · 2023-11-14
>
> We are keen to continue the discussion. Are there any remaining comments following our rebuttal or have we addressed all the reviewer’s concerns?

---

### Official Review · Reviewer_y1v2 · 2023-11-03

**Soundness:** 1 poor
**Presentation:** 1 poor
**Contribution:** 2 fair
**Rating:** 3
**Confidence:** 4

**Summary:**

This work proposes to address deepfake detections by focusing on checking the claimed facts (e.g. identify, motion).

For a potentially faked sample with a claimed fact, they use a pre-trained feature extractor to embed it and compare the embedding to the embedding of a real sample with the same claimed fact. If the cosine similarity between them are high, they predict the sample real; Otherwise they predict the sample as deepfake.

They evaluate this for the cases of detecting face wwapping, Audio-Visual Deepfake and deepfake from text-to-image models.

**Strengths:**

1. Detecting Deepfakes is a hot topic.
2. The proposed method has a very simple formulation.

**Weaknesses:**

**1. Falsely claimed robustness against zero-day attacks.**
The arguments 'the proposed framework is robust against zero-day attacks because it uses pre-trained feature extractor without tuning on deepfake samples' is not justified. It is essentially claiming the following impossibility for generative models: Given a feature extractor, one cannot train a generative model to make the embedding of the generated samples close to a target (real) sample. This is a very strong and likely wrong claim as least for existing approaches.

**2. The limitation/effect of the reference sample (i.e. the real sample with the claimed fact) is not properly investigated and discussed.**
How can one retrieve such a reference sample in practice? How will distribution shifts (e.g. different angles, different cameras, different places) affect the effectiveness of the propose detection method? In most if not all existing experiments, the reference samples come from the same dataset as the real samples, which renders the evaluation less practical. One way to have some preliminary results on these is to apply some augmentations (e.g. rotations, blurring, noises, cropping) to only the reference samples, and see if the performance remains.

**3. The requirement to the feature extractor is over-simplified.** The assumption is basically requiring the pre-trained feature extractor to naturally distinguish real samples from fake ones. This is also a rather strong assumption (it is in fact related to Weakness 1). Please provide some justifications. It is worth noting that at least for now, fooling a pre-trained feature extractor should be considered relatively easier than 'encode the false fact into fake media with sufficient accuracy' given that 1) adversarial robustness remains an open challenge and 2) the pre-trained feature extractor has never seen deepfake samples according to your assumption, which means it may not need to detect 'tiny flaws' when it is trained.

**4. Missing important details in some experiments.** For example: What are method A and method B in section 5? How do you decide the threshold for predicting real/fake? What are the thresholds?

**Questions:**

Please refer to Weakness section above.

---

> ### Author Response · Authors · 2023-11-12
> **Comment to Reviewer y1v2 (1/2)**
>
> We appreciate the reviewer's engagement with our work and address the concerns raised.
>
> **“It is essentially claiming …Given a feature extractor, one cannot train a generative model to make the embedding of the generated samples close to a target (real) sample. This is very strong and likely wrong”.**
> We do not claim that generative models will never be able to create perfectly realistic fake images in the future i.e. with the exact claimed identity and perfectly aligned audio-visual data. We claim (and provide empirical evidence) that current models do not achieve this. We think it may take some time until they do so. Our method is therefore highly effective against current models.
> The reviewer might be suggesting that attacks may cheat by generating target identities that overfit against specific pretrained encoders while not yet solving the difficult task of perfect identity transfer. This is reasonable. Indeed we showed in the CLIP experiment that text-to-image models can overfit to specific backbones well. However, as also shown in the same experiment, it is possible to detect such false facts by using other backbones, different from those used to train the overfitted attack model e.g. as we showed for BLIP. Note that the advantage is on the defenders’ side, as the attacker has no idea which feature extractor FACTOR will use (or indeed an ensemble of feature extractors). See also the response below where we show our face forgery detection is robust to the choice of identity encoder.
>
> **“How can one retrieve such a reference sample in practice? How will distribution shifts (e.g. different angles, different cameras, different places) affect the effectiveness of the proposed detection method?”**
> The reference set in the empirical evaluation uses other videos of the same person. These videos differ in background, camera, angle etc. It therefore already showcases significant resilience to distribution shifts of the type that the reviewer requested.
> In practice, a reference set can be constructed by retrieving public videos or images of the claimed identity. For example, if we claim a particular video is of Obama, we would go to Google Image Search and retrieve a set of images of Obama. This will not only work for past US presidents like Obama, but also for ordinary people that have some media presence e.g. a Facebook or Linkedin profile image, previously shared Instagram photos or Youtube videos. Of course, if the claimed identity has no known images, our method does not apply. However, as around 2 billion people have Facebook profiles, we believe that our method is useful in many cases.
>
> **“The assumption is basically requiring the pre-trained feature extractor to naturally distinguish real samples from fake ones … fooling a pre-trained feature extractor should be considered relatively easier than 'encode the false fact into fake media with sufficient accuracy'”.**
> We do not assume that pretrained features can naturally distinguish real and fake images, but rather they can help recognize if the claimed fact is true or not. For example, if the identity is truly the claimed one or the audio and visual data in correspondence. This is the core of our approach which led to state-of-the-art results in detection of zero-day deepfake attacks.
> Note that we (and most other deepfake detection works) do not tackle the adversarial case. As the defender is free to choose any backbone they see fit, it would not be easy for the attacker to know which backbone to attack, and it is even harder to be adversarial to all possible backbones simultaneously. Also, many face swapping attacks we tested against do include deep pretrained features of the target identity therefore being somewhat adversarial to our method. The results showed that our method was still effective against such attacks. Furthermore, our method is not overfitted to a single feature extractor. We found that CLIP, which has been trained on quite irrelevant data, is already quite effective. We included the results here and in the revised manuscript.
>
> $\text{Dataset}\hspace{0.75cm}\text{CLIP}\hspace{0.5cm}\text{Swin-S}\hspace{0.5cm}\text{Attention-92(MX)}$
>
> $\text{Celeb-DF}\hspace{0.6cm}89.3\hspace{0.625cm}94.5\hspace{1.55cm}\textbf{97.0}$
>
> $\text{DFD}\hspace{1.45cm}95.7\hspace{0.625cm}96.1\hspace{1.55cm}\textbf{96.3}$
>
> $\text{DFDC}\hspace{1.15cm}98.8\hspace{0.625cm}99.3\hspace{1.55cm}\textbf{99.9}$

---

> ### Author Response · Authors · 2023-11-12
> **Comment to Reviewer y1v2 (2/2)**
>
> **“What are method A and method B in section 5?”.** Method A and method B are terms used in Sec.5 to denote the deepfake generation methods within the DFDC dataset. Note that the dataset does not provide specific information about the technical details of these deepfake generation techniques; they are simply identifiers within the DFDC dataset metadata. We clarified this in the revision.
>
> **“How do you decide the threshold for predicting real/fake? What are the thresholds?”.** As we share the reviewer’s worries about thresholds, our evaluation metric does not use thresholds at all! We use AUROC (Area Under the Receiver Operating Characteristic curve) and AUPR (Area Under the Precision-Recall curve), which measure performance across various operating points without relying on predefined thresholds. Most other deepfake detection papers use the same evaluation approach.

---

> ### Author Response · Authors · 2023-11-14
>
> We are keen to continue the discussion. Are there any remaining comments following our rebuttal or have we addressed all the reviewer’s concerns?

---

> > ### Comment · Reviewer_y1v2 · 2023-11-14
> >
> > Weakness 1, 2 and 3 in my review are not addressed, among which weakness 1 and weakness 3 are the most critical issues.
> >
> > >**Weakness 1. Falsely claimed robustness against zero-day attacks.** The arguments 'the proposed framework is robust against zero-day attacks because it uses pre-trained feature extractor without tuning on deepfake samples' is not justified. It is essentially claiming the following impossibility for generative models: Given a feature extractor, one cannot train a generative model to make the embedding of the generated samples close to a target (real) sample. This is a very strong and likely wrong claim as least for existing approaches.
> >
> > To my understanding, the rebuttal basically confirm this issue. If not, could the authors clarify **in what sense** should the proposed scheme be robust against zero-day attacks? It might help if the claim can be made with somewhat formal descriptions.
> >
> > >**Weakness 3. The requirement to the feature extractor is over-simplified.** The assumption is basically requiring the pre-trained feature extractor to naturally distinguish real samples from fake ones. This is also a rather strong assumption (it is in fact related to Weakness 1). Please provide some justifications. It is worth noting that at least for now, fooling a pre-trained feature extractor should be considered relatively easier than 'encode the false fact into fake media with sufficient accuracy' given that 1) adversarial robustness remains an open challenge and 2) the pre-trained feature extractor has never seen deepfake samples according to your assumption, which means it may not need to detect 'tiny flaws' when it is trained.
> >
> > While this assumption is never explicitly made, the effectiveness of the proposed approach indeed requires that the pre-trained feature extractor naturally distinguishes real samples from fake ones, doesn't it? The issue is not just about adversarial cases. Speaking from a high level, the proposed approach basically reduce (with assumption that there is an explicitly claimed fact) the problem of deepfake detections to the new problem of differentiating real/faked claimed fact. However, this reduction might not be very meaningful if the problem of differentiating real/faked claimed fact is not easy and simply assuming a pre-trained feature extractor can handle the new problem is misleading and not responsible.
> >
> >
> > >**Weakness 2. The limitation/effect of the reference sample (i.e. the real sample with the claimed fact) is not properly investigated and discussed.** How can one retrieve such a reference sample in practice? How will distribution shifts (e.g. different angles, different cameras, different places) affect the effectiveness of the propose detection method? In most if not all existing experiments, the reference samples come from the same dataset as the real samples, which renders the evaluation less practical. One way to have some preliminary results on these is to apply some augmentations (e.g. rotations, blurring, noises, cropping) to only the reference samples, and see if the performance remains.
> >
> > This is still important (though it is less critical compared to Weakness 1 and 3). Please take a closer look at my original comments since they seem to be overlooked given the rebuttal.
> >
> >
> >
> > Overall, I think the high-level idea of fact checking as a tool to mitigate at least some threats from deepfake is very promising and very interesting. However, the current solution incorporating this idea is ill-supported and likely flawed. I would suggest authors to focus on the explicit/implicit assumptions of their approach.

---

> > > ### Author Response · Authors · 2023-11-15
> > >
> > > Thank you for your reply. Here are our clarifications.
> > >
> > > **Pretrained feature extractors:** We now understand the confusion regarding pretrained feature extractors. Our approach doesn’t require the use of pretrained feature extractors. In cases where a sufficiently large number of real images and fact labels are available, we can train an effective feature extractor. However, given the relatively small size of current deepfake datasets, we find it more practical to leverage available pretrained feature extractors pretrained on large auxiliary datasets. We will elucidate this point in the text to avoid further confusion.
> > >
> > > **Zero-day attacks:** Being robust to zero-day attacks is generalizing to fake images resulting from attacks not seen during training. By not exposing to deepfake images in training, all deepfake attacks are esssentially zero-day attack to our method. This is different from methods that train a classifier between real and fake data, where the attacks seen in training are not zero-day. The results in our paper indeed demonstrate the robustness of our approach to zero-day attacks.
> > >
> > > **Fact checking is sometimes difficult:** Not all facts are easy to check. We have shown two important examples where facts are reasonably easy to check, and we are certain that there are other scenarios where facts are easily verifiable. Conversely, there are facts that are very difficult to verify, such as the exact time and location where an image was taken. In this case fact-checking may not be helpful. As face-swapping and audio-visual attacks are very important, they already demonstrated the significance of our approach.
> > >
> > > **Investigation of reference samples:** In our response, we highlighted that the reference set used in our empirical evaluation comprises videos showcasing the same person with variations in background, camera angle, face rotation, cropping, video blurs, JPEG compressions, noises, etc. This demonstrates that our approach is robust to distribution shifts.
> > >
> > > We appreciate that the reviewer acknowledged fact-checking as a promising avenue to mitigate deepfake threats. We hope these clarifications address the reviewer’s concerns, and we remain open to further discussion or elaboration on any specific points.

---

### Author Response · Authors · 2023-11-12
**General Comment**

We thank all reviewers for the effort invested in their reviews. We responded to each reviewer individually, we also uploaded a revised version of the manuscript and the code of our method (including train/test splits). We are more than happy to engage in discussion.